# Increase in Cape Verde hurricanes during Atlantic Niño

Dongmin Kim [1,2] ✉, Sang-Ki Lee [2], Hosmay Lopez [2], Gregory R. Foltz[2], Caihong Wen[3], Robert West[2,4] & Jason Dunion[1,2]

At seasonal-to-interannual timescales, Atlantic hurricane activity is greatly modulated by El Niño–Southern Oscillation and the Atlantic Meridional Mode. However, those climate modes develop predominantly in boreal winter or spring and are weaker during the Atlantic hurricane season (June–November). The leading mode of tropical Atlantic sea surface temperature (SST) variability during the Atlantic hurricane season is Atlantic Niño/Niña, which is characterized by warm/cold SST anomalies in the eastern equatorial Atlantic. However, the linkage between Atlantic Niño/Niña and hurricane activity has not been examined. Here, we use observations to show that Atlantic Niño, by strengthening the Atlantic inter-tropical convergence zone rainband, enhances African easterly wave activity and low-level cyclonic vorticity across the deep tropical eastern North Atlantic. We show that such conditions increase the likelihood of powerful hurricanes developing in the deep tropics near the Cape Verde islands, elevating the risk of major hurricanes impacting the Caribbean islands and the U.S.

Seasonal outlooks for Atlantic tropical cyclone (TC) activity are issued by the National Oceanic and Atmospheric Administration (NOAA), the European Center for Medium-Range Weather Forecasts and other agencies to provide a general guide to the expected overall activity during the upcoming hurricane season. These outlooks are largely based on foundational relationships between observed oceanic and atmospheric states. For instance, during La Niña, the negative phase of El Niño–Southern Oscillation (ENSO), cold equatorial Pacific sea surface temperature anomalies (SSTAs) produce a fast tropical teleconnection (via atmospheric Kelvin waves) that decreases the atmospheric static stability and vertical wind shear over the TC main development region (MDR) and thus increase Atlantic TC activity[1,2]. The positive phase of the Atlantic Meridional Mode (AMM), characterized by warm SSTAs and low-level westerly wind anomalies over the tropical North Atlantic, also decreases the atmospheric static stability and vertical wind shear over the MDR and thus increases Atlantic TC genesis[3]. Conversely, El Niño, the positive phase of ENSO, and the negative phase of AMM both tend to increase the atmospheric static

stability and vertical wind shear over the MDR and thus suppress Atlantic TC genesis[4–8].

Seasonal TC activity in the Atlantic basin is also closely tied to the West African summer monsoon intensity. Specifically, the West African summer monsoon promotes synoptic-scale low-pressure disturbances known as African Easterly Waves (AEWs) that often provide the mechanical uplift of surface air parcels above the level of free convection and thus can trigger TC genesis[9–16]. The West African summer monsoon is also directly connected to the seasonal development of the West African westerly jet in the deep tropical eastern North Atlantic centered around 10°N[17,18]. The westerly jet produces positive low-level relative vorticity (i.e., low-level cyclonic rotation) along its northern flank, promoting the formation of TCs in the deep tropical eastern North Atlantic[17].

These climate modes of variability and the associated atmosphere-ocean processes are well-established predictors or indicators for seasonal Atlantic TC activity. While the use of potentially predictable climate modes has made skillful seasonal Atlantic TC

[1]Cooperative Institute for Marine and Atmospheric Studies, University of Miami, Miami, FL, USA. [2]Atlantic Oceanographic and Meteorological Laboratory, NOAA, Miami, FL, USA. [3]Climate Prediction Center, NOAA, College Park, MD, USA. [4]Northern Gulf Institute, Mississippi State University, Starkville, MS, USA. ✉e-mail: dongmin.kim@noaa.gov

outlooks possible, there is still much uncertainty in these forecasts. For instance, the 2008 Atlantic hurricane season was one of the deadliest and costliest Atlantic hurricane seasons, with 16 named storms and five major hurricanes (i.e., category 3-5 hurricanes based on the Saffir-Simpson hurricane wind scale) including a powerful category-4 hurricane (Ike) that resulted in 164 casualties and $30 billion damage in the Caribbean islands and the U.S[19]. During the 2008 Atlantic hurricane season, however, both ENSO and AMM remained in neutral conditions. Coincidentally, the equatorial Atlantic was dominated by Atlantic Niño conditions, typically characterized by warm SSTAs and deeper thermocline in the eastern basin and low-level westerly wind anomalies in the western basin[20–22].

Similarly, during the 2003 hurricane season, Atlantic Niño conditions developed in the eastern equatorial Atlantic as early as July and continued until January 2004, while both ENSO and AMM remained in near-neutral conditions. The 2003 hurricane season was highly active with 16 named storms, including three major hurricanes. Among those, a tropical depression developed near the Cape Verde islands in early September, triggered by a strong AEW from the West African coast. It later strengthened to a powerful category-5 hurricane (Isabel), made landfall in North Carolina, and resulted in 51 fatalities and $3.37 billion in damage[19]. Conversely, the 1992 hurricane season had only four named storms with an Accumulated Cyclone Energy of $76.2 \times 10^4$ kt$^2$,

both significantly below average[19]. During that season, ENSO and AMM were both in neutral phases, while Atlantic Niña conditions developed, characterized by cold eastern equatorial Atlantic SSTAs and low-level easterly wind anomalies in the western equatorial Atlantic. These cases during 1992, 2003, and 2008 suggest that Atlantic Niño/Niña is potentially linked to Atlantic TC activity.

Previous studies have shown that Atlantic Niño/Niña strongly modulates the inter-tropical convergence zone (ITCZ) rainband, with the ITCZ and sub-Sahel West African rainfall enhanced during Atlantic Niño and suppressed during Atlantic Niña[23–30]. Based on the well-established relationships between Atlantic Niño/Niña, the ITCZ and the West African rainfall, we hypothesize that Atlantic Niño/Niña modulates Atlantic TC activity through its impact on atmospheric convection from the deep tropical eastern North Atlantic to sub-Sahel West Africa and the associated changes in regional atmospheric circulations and AEW activity. Consistent with this hypothesis, the number of TCs generated over the tropical North Atlantic (60°W–10°W, 5°N–20°N) is significantly higher during Atlantic Niño (5.1 ± 0.58 TC/year) and lower during Atlantic Niña (2.9 ± 0.61 TC/year) (Fig. 1a). This difference is significant at the 95% confidence level based on a two-tailed Student's $t$-test and thus supports our hypothesis that eastern equatorial Atlantic SSTAs linked to Atlantic Niño/Niña influence Atlantic TC activity.

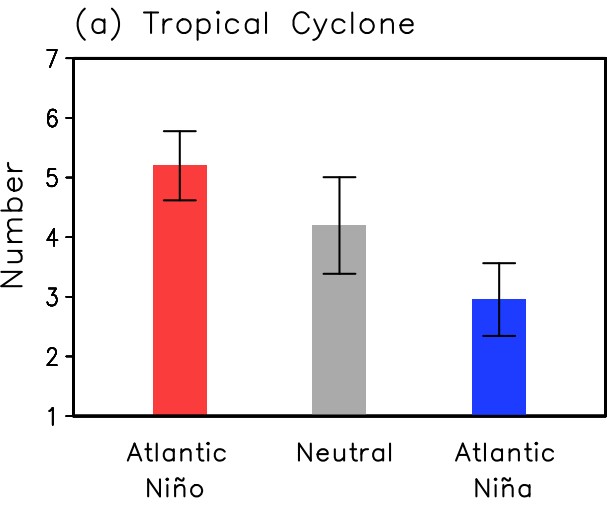

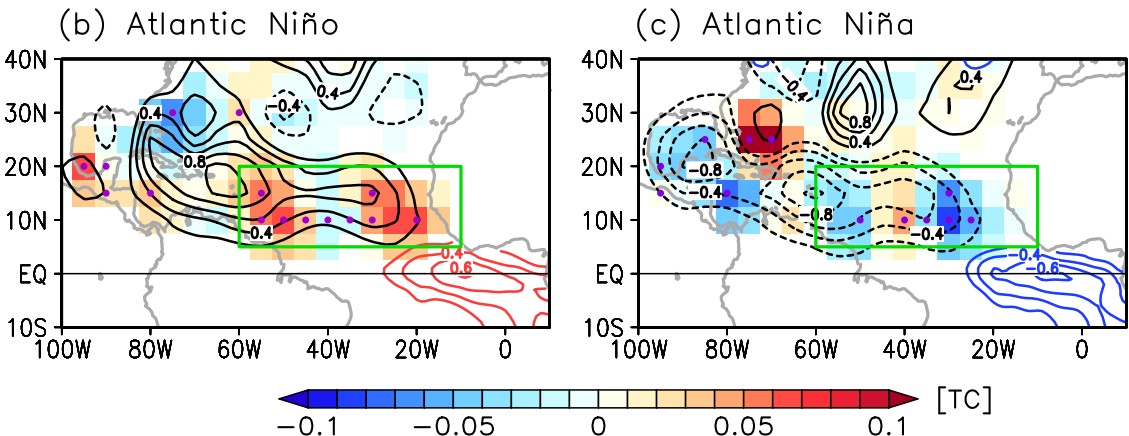

**Fig. 1 | Composite analysis of tropical cyclone activity in the tropical North Atlantic during Atlantic Nino/Nina. a** Number of June–November tropical cyclones (TC) over the tropical North Atlantic (60°W–10°W, 5°N–20°N; green boxes in (**b**) and (**c**)) during Atlantic Niño (red bar), neutral (gray bar) and Atlantic Niña (blue bar) years. The error bars indicate the 95% confidence level based on a two-tailed Student's $t$-test. Spatial patterns of anomalous tropical cyclone genesis (shaded, per year), track density (black contours, intervals of 0.2 TCs per year), and sea surface temperature (red and blue contours, intervals of 0.2 K) composites during **b** Atlantic Niño and **c** Atlantic Niña. Purple dots indicate where TC genesis anomalies are significant above the 95% confidence level based on a Student's $t$-test. TC genesis and track density are spatially smoothed to aid visual comparison.

The main objectives of this study are to investigate further the links between Atlantic Niño/Niña and seasonal Atlantic TC activity and explore the associated physical mechanisms by analyzing observational and reanalysis datasets. We show the critical role of Atlantic Niño/Niña in the development of major Atlantic hurricanes in the deep tropics near the Cape Verde islands.

## Results

### Atlantic TC genesis and track density during Atlantic Niño/Niña

The spatial patterns of Atlantic TC genesis and track density linked to Atlantic Niño and Niña are first examined using a composite analysis for 74 years (1948–2021; Fig. 1b, c). During Atlantic Niño, TC genesis increases over the Caribbean Sea and tropical North Atlantic, especially the eastern tropical North Atlantic (east of 40°W), while TC genesis decreases over the western North Atlantic between 20°N and 30°N. The increased frequency of TC genesis over the Caribbean Sea and tropical North Atlantic increases TC track density, elevating the probability of TC landfall in the Caribbean islands and around Florida. In contrast, during Atlantic Niña, Atlantic TC genesis decreases over the Gulf of Mexico, Caribbean Sea, and tropical North Atlantic and slightly increases over the western North Atlantic between 20°N and 30°N, resulting in a decrease in TC track density over the Gulf of Mexico, Caribbean Sea, and tropical North Atlantic.

Two leading modes of SSTA variability in the tropical Atlantic are Atlantic Niño/Niña and AMM. Although some Atlantic Niño/Niña events are preceded by AMM events in boreal spring[29,31–34], the correlation between ATL3 and AMM indices (Methods) during the Atlantic hurricane season (JJASON) is virtually zero ($r = -0.03$). However, while the correlation between ATL3 and NIÑO3.4 indices in JJASON is statistically insignificant based on a two-tailed Student's t-test ($r = -0.15$), some Atlantic Niño/Niña events are directly forced by ENSO in boreal winter and spring, and Atlantic Niño/Niña may influence ENSO evolution in boreal summer and fall[30,35–38]. Therefore, to properly address the potential interbasin interactions and their impact on Atlantic TC activity, it is necessary to separate the impact of Atlantic Niño/Niña on Atlantic TCs from that of ENSO. To do so, we carried out partial regressions of Atlantic TC genesis and track density onto ATL3 and NIÑO3.4 indices (Fig. 2a, b). Note that the sign of NIÑO3.4 is reversed in Fig. 2b to help visual comparison. As shown in Fig. 2a, Atlantic Niño leads to an increase in Atlantic TC genesis, almost exclusively over the eastern tropical North Atlantic (east of 40°W). The elevated TC activity there leads to an increase in TC track density across the Caribbean islands and around Florida. On the other hand, La Niña tends to increase TC genesis predominantly in the southern Gulf of Mexico and Caribbean Sea, and to increase track density across Central America, Caribbean islands, and the southern and southeastern U.S. (Fig. 2b). These results are largely consistent and more robust during the post-satellite period of 1979-2021 (Supplementary Fig. S1). Next, we explore the physical mechanisms through which Atlantic Niño/Niña and ENSO modulate Atlantic TC activity.

### Atlantic Niño/Niña and atmospheric TC environments

During Atlantic Niño, the ITCZ rainband strengthens in the deep tropical North Atlantic around 0°N–10°N and 30°W–10°E, enhancing sub-Sahel West African rainfall and regional low-level winds (Fig. 2c)[23,25,29,30,39–41]. The enhanced ITCZ and sub-Sahel rainfall directly reinforce AEW activity in the deep tropics across the eastern tropical North Atlantic (Fig. 2c)[42]. Additionally, the enhanced ITCZ produces low-level westerly wind anomalies (Supplementary Fig. S2a) over the tropical North Atlantic. This, in turn, enhances the West African westerly jet and thus strengthens positive low-level relative vorticity in the eastern tropical North Atlantic between 5°N and 15°N (Fig. 2e). In the western tropical North Atlantic (west of 40°W) away from the West African westerly jet, the vertical wind shear is reduced (Fig. 2e) due to weakening of both the upper-level westerly and low-level easterly wind

in response to the enhanced ITCZ (Supplementary Fig. S2a, c). These large-scale atmospheric patterns increase the likelihood of powerful TCs forming in the deep tropics near the Cape Verde islands (i.e., Cape Verde hurricanes), and the probability of strengthening as they move westward toward the Caribbean islands and the area around Florida (Fig. 2a).

During La Niña, a fast tropical teleconnection from the equatorial Pacific produces an increase in atmospheric convection mainly over the tropical North Atlantic west of 30°W (Fig. 2d). AEW activity is enhanced near the West African coast around 15°–20°N, but is largely suppressed away from the coast (Fig. 2d). La Niña tends to decrease vertical wind shear and produce positive low-level relative vorticity anomalies predominantly in the southern Gulf of Mexico and Caribbean Sea, and thus results in an increase in TC activity therein (Fig. 2b, f), consistent with previous studies[4,43–48].

### Impact of Atlantic Niño/Niña on major hurricanes

Figure 3a shows the spatial distributions of total TC genesis (contours) and major hurricane genesis (shaded) during the 74 years of observed records (1948–2021). While TC genesis is widespread over the tropical North Atlantic, Caribbean Sea, Gulf of Mexico, and western North Atlantic, the formation of major hurricanes is particularly strong in the deep tropics over the eastern tropical North Atlantic (east of 40°W). Compared to TCs generated over the Gulf of Mexico and the Caribbean Sea, TCs that form in the deep tropics over the eastern tropical North Atlantic can travel farther over the warm ocean, giving them more time to build moist energy before they affect the Caribbean islands, Central America, and the U.S. Since Atlantic Niño/Niña modulates the formation of TCs in the deep tropics near the Cape Verde islands (Fig. 2a), it is reasonable to further hypothesize that Atlantic Niño/Niña is linked to major hurricanes.

Consistent with this hypothesis, the number of major hurricanes generated over the tropical North Atlantic (60°W–10°W, 5°N–20°N) is significantly higher during Atlantic Niño years ($2.16 \pm 0.56$ per year) and lower during Atlantic Niña years ($0.92 \pm 0.51$ per year). This difference is significant at the 95% confidence level based on a two-tailed Student's t-test and is stronger during the post-satellite period (i.e., Atlantic Niño: $2.5 \pm 0.71$ per year versus Atlantic Niña: $0.2 \pm 0.57$ per year in Supplementary Fig. S3). In contrast, the number of major hurricanes that form over the tropical North Atlantic in El Niño years ($1.22 \pm 0.52$ per year) is not statistically different, at the 95% confidence level, from the number that forms during La Niña years ($1.85 \pm 0.58$ per year). This result is largely consistent and more robust during the post-satellite period (Supplementary Fig. S3).

Figure 3c–e shows correlation maps of major hurricane genesis from observations (based on IBTrACS, see Methods) and the reconstructions based on the NIÑO3.4 index (Fig. 3c), ATL3 index (Fig. 3d), and both NIÑO3.4 and ATL3 indices (Fig. 3e). The reconstructed major hurricane genesis using only the NIÑO3.4 index shows good agreement with IBTrACS over the Caribbean Sea and central tropical North Atlantic around 10°–20°N and 40°–50°W (Fig. 3c). However, the NIÑO3.4 index provides poor skill in reproducing major hurricane genesis over the deep tropical eastern North Atlantic around 5°–15°N and 40°–15°W. In contrast, the reconstructed major hurricane genesis using the ATL3 index represents major hurricane genesis substantially better over the eastern tropical North Atlantic (east of 40°W) compared to that derived from the NIÑO3.4 index (Fig. 3d). Therefore, the reconstructed major hurricane genesis derived from the NIÑO3.4 and ATL3 indices represents major hurricane genesis in the North Atlantic substantially better than that derived from the individual indices (Fig. 3e).

## Discussion

In this study, we show that Atlantic Niño/Niña produces atmospheric conditions conducive to the formation of TCs in the deep tropics near

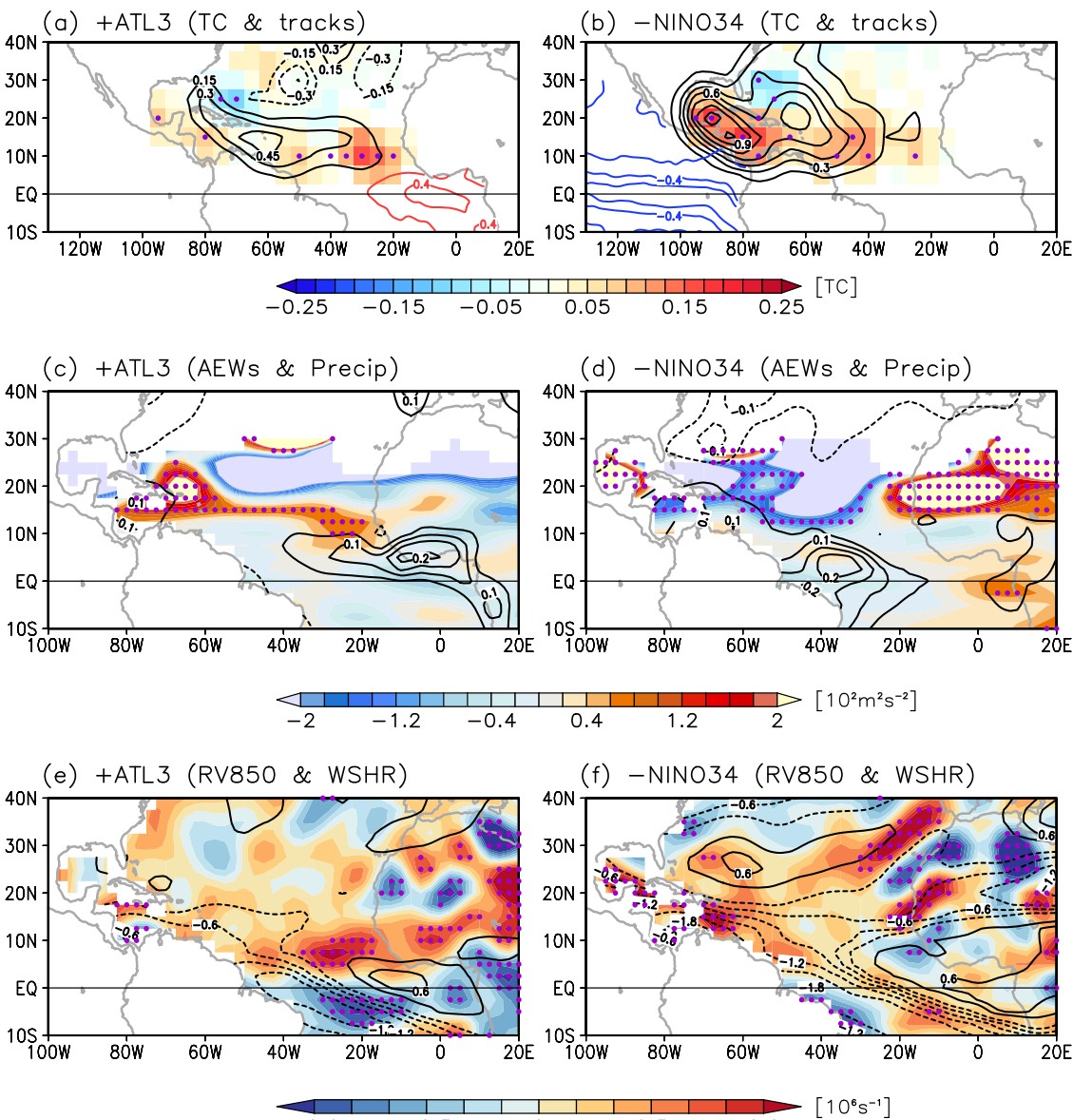

**Fig. 2 | Impact of Atlantic Nino/Nina and El Niño–Southern Oscillation on Atlantic tropical cyclone activity and associated large-scale environment.** Partial regressions of tropical cyclone genesis (shaded), track density (black contours), and sea surface temperature anomalies (red and blue contours, interval is 0.2 K) onto **a** Atlantic Niño (ATL3) and **b** Niño 3.4 (NINO3.4) indices. **c** and **d** are similar to (**a**) and (**b**) but for eddy kinetic energy at 700 hPa (shaded, $10^2$ m$^2$ s$^{-2}$) as a measure of African easterly waves (AEWs) and precipitation (contours, mm day$^{-1}$). **e** and **f** are similar to (**a**) and (**b**) but for low-level relative vorticity at 850 hPa (RV850, shaded, $10^6$ s$^{-1}$) and vertical wind shear between 850 and 200 hPa (WSHR, contours, m s$^{-1}$). Purple dots indicate where regressions of TC genesis (in (**a**) and (**b**)), eddy kinetic energy (in (**c**) and (**d**)), and low-level relative vorticity anomalies (in (**e**) and (**f**)) are significant above the 95% confidence level based on a Student's *t*-test. Note that the sign of NINO3.4 is reversed, and the regressed values of eddy kinetic energy, precipitation, low-level relative vorticity and vertical wind shear over the Americas and Pacific Ocean are masked out. In addition, the area with climatologically westerly winds in (**c**) and (**d**) is masked out.

the West African coast, the breeding ground for powerful Cape Verde hurricanes. More specifically, warm eastern equatorial Atlantic SSTAs associated with Atlantic Niño strengthen the ITCZ rainband, enhancing sub-Sahel West African rainfall and AEW activity (i.e., increased EKE) in the deep tropics (5°–15°N) east of 40°W. The enhanced ITCZ also reinforces the West African westerly jet, strengthening positive low-level relative vorticity in the deep tropical eastern tropical North Atlantic. These atmospheric anomalies lead to an increase in the formation of deep tropical Atlantic TCs in the eastern tropical North Atlantic (i.e., Cape Verde hurricanes) and thus potentially increase the probability of TC landfall in the Caribbean islands and the area around Florida (Fig. 4a). Since TCs that form in the deep tropics near the Cape Verde islands are more likely to develop into major hurricanes

(Fig. 3a)[49], Atlantic Niño/Niña is significantly linked to the number of major hurricanes generated over the tropical North Atlantic. This relationship is even stronger during the post-satellite period, suggesting that not all major hurricanes generated in the tropical North Atlantic are accounted for in IBTrACS during the pre-satellite period[50]. The influence of Atlantic Niño/Niña on Atlantic TC activity is quite different from that of ENSO, which is predominantly linked to TCs that form in the southern Gulf of Mexico, Caribbean Sea, and central tropical North Atlantic (Figs. 2b and 3c).

Although Atlantic Niño/Niña was found to significantly influence the genesis of Cape Verde hurricanes, the overall seasonal Atlantic TC activity is still determined predominantly by ENSO and AMM (Supplementary Fig. S4). Therefore, it is more likely that Atlantic Niño/Niña

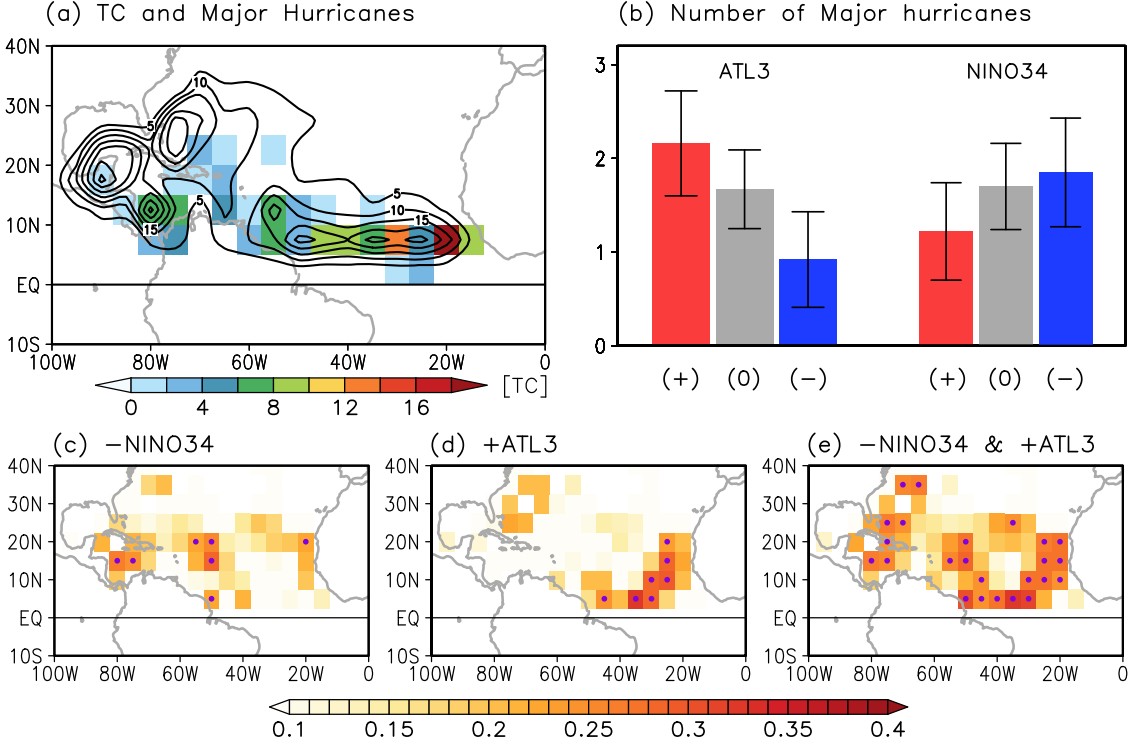

**Fig. 3 | Impact of Atlantic Niño/Niña and El Niño–Southern Oscillation on the genesis of major hurricanes in the tropical North Atlantic. a** Spatial patterns of tropical cyclone (TC) genesis for all-category (contours, total number of TC genesis, intervals of 5 TC per contour line) and category 3-5 TCs (shaded, total number of category 3-5 hurricane genesis) for 74 years (1948–2021). **b** The number of major hurricanes (category 3-5) formed over the tropical North Atlantic (60°W–10°W, 5°N–20°N) during positive (red bars), neutral (gray bars), and negative (blue bars) phases of (left) Atlantic Niño (ATL3) and (right) Niño 3.4 (NINO34) indices. The error bars indicate the 95% confidence level based on a two-tailed Student's t-test. Correlation maps of category 3-5 hurricane genesis between observations and reconstructions derived from partial linear regressions of **c** NINO3.4 index, **d** ATL3 index, and **e** their sum. Purple dots in (**c**)–(**e**) indicate where correlations are significant above the 95% confidence level based on a two-tailed Student's t-test.

modulates the locations of TC genesis and tracks while the overall seasonal Atlantic TC activity is largely determined by ENSO and AMM. Nevertheless, Atlantic TC activity can be enhanced or suppressed when Atlantic Niño/Niña and ENSO are out of phase (i.e., La Niña & Atlantic Niño conditions, and El Niño & Atlantic Niña conditions), and weakly influenced when they are in phase. Similarly, Atlantic TC activity can be enhanced or suppressed when Atlantic Niño/Niña and AMM are in phase (i.e., positive AMM & Atlantic Niño conditions, and negative AMM & Atlantic Niña conditions), and weakly influenced when they are out of phase (i.e., positive AMM & Atlantic Niña conditions, and negative AMM & Atlantic Niño conditions). These can be clearly seen in both partial regression and composite analysis results (Supplementary Figs. S5 and S6).

Finally, Atlantic Niño/Niña is a potentially predictable climate mode of variability that prevails during the Atlantic hurricane season. It is also statistically independent from ENSO ($r = -0.15$) and AMM ($r = -0.03$), two well-established predictors of Atlantic seasonal TC activity. As such, Atlantic Niño/Niña may serve as an additional predictor to improve seasonal Atlantic hurricane outlooks, especially when ENSO and AMM are in near-neutral phases, as occurred during the 1992, 2003, and 2008 hurricane seasons.

## Methods
The International Best Track Archive for Climate Stewardship (IBTrACS)[51] is used to derive Atlantic TC genesis and track density. We only consider TCs with wind speeds greater than 34 knots; thus, tropical depressions are excluded. The National Centers for Environmental Prediction–National Center for Atmospheric Research Reanalysis version 1 (NCEP1)[52] is used to derive monthly atmospheric circulation fields. Monthly precipitation is obtained from NOAA's

Gridded Precipitation Reconstruction dataset[53]. SST is derived from the Hadley Centre Global Sea Ice and Sea Surface Temperature version 1 (HadISST1)[54]. All analyses carried out in this study are for the Atlantic hurricane season (June–November; JJASON) spanning 74 years (1948–2021). The results are largely consistent and more robust during the post-satellite period (Supplementary Fig. S3). All data were linearly detrended to avoid the potential influence of anthropogenic climate change. Atlantic Niño and Niña are identified based on the criterion that the area-averaged SSTAs over the ATL3 region (3°S–3°N, 20°W–0°)[21] exceed one standard deviation ($\sigma = 0.32$ K) during JJASON or are less than negative one standard deviation, respectively. In total, 19 Atlantic Niño and 14 Atlantic Niña events were identified (Supplementary Table S1). The same approach was followed to identify El Niño and La Niña events ($\sigma = 0.50$ K), but using area-averaged SSTAs over the NIÑO3.4 region (5°S–5°N, 170°E–120°W). In total, 17 El Niño and 21 La Niña events were identified (Supplementary Table S1) over the 74-year study period. Similarly, positive and negative AMM years are identified when the normalized AMM index during JJASON exceeds one standard deviation (positive AMM) or is less than negative one standard deviation (negative AMM). In total, 15 positive AMM years and 18 negative AMM years were identified (Supplementary Table S1).

Eddy kinetic energy (EKE) at 700 hPa is often used as a measure of AEW activity (Eq. 1)[16]. Following a previous study[55], we calculate EKE as

$$\text{EKE} = \frac{1}{2}\left(u'^2 + v'^2\right), \tag{1}$$

where $u'$ and $v'$ indicate the zonal and meridional eddy wind components, respectively, defined using a wavenumber-frequency filter that

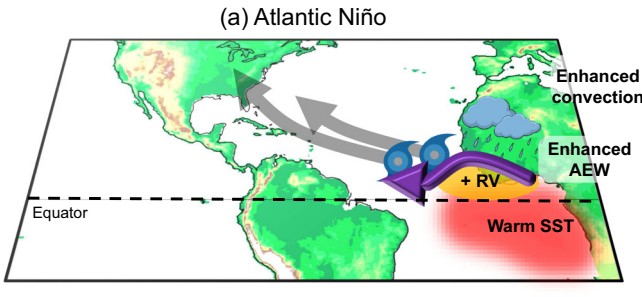

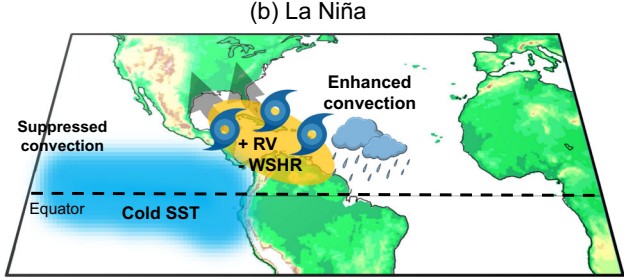

**Fig. 4 | Schematic of the impact of Atlantic Niño and La Niña on Atlantic tropical cyclone activity. a** A summary schematic illustrates the impact of **a** Atlantic Niño and **b** La Niña on Atlantic tropical cyclone (TC) activity. Blue and red shades indicate cold and warm sea surface temperature (SST) anomalies, respectively. The purple arrow in (**a**) indicates the African easterly waves (AEW). Orange shades in (**a**) represent positive low-level relative vorticity anomalies (RV), and both positive low-level relative vorticity anomalies and negative vertical wind shear (WSHR) anomalies in (**b**). Thick gray arrows represent TC tracks.

retains periods of 2–10 days and wavenumbers of 10°–40° longitude (approximately 1000–4000 km equivalent wavelength).

In order to objectively separate the impact of Atlantic Niño/Niña on Atlantic TC from that of ENSO, we computed partial regressions of Atlantic TC genesis and track density onto ATL3 and NIÑO3.4 indices as follows

$$TC_{ENSO\_ATL3}(t,x,y) = \beta_1(x,y) \cdot NINO34(t) + \beta_2(x,y) \cdot ATL3(t) \quad (2)$$

$$TC_{ENSO}(t,x,y) = \beta_1(x,y) \cdot NINO34(t) \quad (3)$$

$$TC_{ATL3}(t,x,y) = \beta_2(x,y) \cdot ATL3(t) \quad (4)$$

where the $TC_{ENSO\_ATL3}$ is the reconstructed TC variable using partial regressions (Eq. 2, Fig. 3e). $\beta_1$ is the partial regression coefficient of NIÑO3.4. It represents TC change per unit change in NIÑO3.4 while ATL3 is held constant. Similarly, $\beta_2$ is the partial regression coefficient of ATL3, representing TC change per unit change in ATL3 while NIÑO3.4 is held constant. NÑO3.4 and ATL3 are normalized. $TC_{ENSO}$ is the reconstructed TC variable using only the partial regression coefficient of NIÑO3.4 (Eq. 3). $TC_{ATL3}$ is same as $TC_{ENSO(t,x,y)}$ but for using only the partial regression coefficient of ATL3 (Eq. 4). Using the reconstructed TC genesis and track density, we computed the correlations between observed and reconstructed TC genesis and track density as shown in Fig. 3c–e. The significance test conducted in this study is based on a standard two-tailed Student's *t*-test.

## Data availability

The National Centers for Environmental Prediction–National Center for Atmospheric Research Reanalysis version 1 (NCEP1) data were downloaded from NOAA PSL at https://psl.noaa.gov/data/gridded/data.ncep.reanalysis.html. Hadley Centre Sea Ice and Sea Surface Temperature (HadISST) data were downloaded from Met Office

Hadley Centre at https://www.metoffice.gov.uk/hadobs/hadisst/. Tropical cyclone data from the International Best Track Archive for Climate Stewardship (IBTrACS) were downloaded from NOAA's National Centers for Environmental Information (NCEI) at https://www.ncei.noaa.gov/products/international-best-track-archive.

## Code availability

All statistical analyses were performed using the Grid Analysis and Display System (GrADS), which is publicly available from the Center for Ocean-Land-Atmosphere Studies at http://cola.gmu.edu/grads and NCL, which is publicly available from the NCAR Command Language (NCL) at https://www.ncl.ucar.edu/ The GrADS, NCL, and Fortran codes used to perform the analyses can be accessed upon request to D.K.

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

## Acknowledgements

This work was carried out under the auspices of the Cooperative Institute for Marine and Atmospheric Studies (CIMAS), a cooperative institute of the University of Miami, and NOAA, cooperative agreement NA20OAR4320472, and supported by NOAA's Climate Program Office, Climate Variability and Predictability Program, and NOAA's Atlantic Oceanographic and Meteorological Laboratory.

## Author contributions

D.K. and S.-K.L. conceived the study. D.K. performed the analysis and wrote the initial draft of the paper. All authors (D.K., S.-K.L., H.L., G.R.F., C.W., R.W. and J.D.) significantly contributed to the discussion and interpretation of results and reviewed and edited the paper.

## Competing interests

The authors declare no competing interests.
