## [Peer Review File · Nature Communications]

Increase in Cape Verde hurricanes during Atlantic NiñoREVIEWER COMMENTS

Reviewer #1 (Remarks to the Author):

Review of "Increase in Cape Verde hurricanes during Atlantic Niño"

The study uses observational data sets 1948-2021 to show that the Atlantic Niño events are linked to tropical cyclones and major hurricanes that originate around Cape Verde off the West African Coast. They authors argue that the Atlantic Niños can be linked to stronger the Atlantic inter-tropical convergence zone and enhanced African easterly wave activities and low-level cyclonic vorticity across the deep tropical eastern North Atlantic, the region where the hurricanes originate. In my assessment, this is a novel and timely analysis that is worth publishing in Nature Communications. The major concern I have is that the conclusions are based largely on partial regressions (of the Atlantic Niño, Atlantic meridional mode and El Niño indices) and more details are needed about this method and the robustness of the results. Are the findings/conclusions here sustained using methods such as composite analysis that simply show the years when the Atlantic Niño occurs alone or together with the other modes? I have a few specific comments below.

L26-28: Not clear what "they" refers to, also when the hurricane season is.

L54-62: The argumentation here is a bit hard to follow; it reads confusing and perhaps even confusing.

L109-114: I think that what is needed here is a summary of what was done and the most important findings.

L117-125: GoM, CBN etc. These unfamiliar acronyms make reading the manuscript difficult, especially where they are not used many time. The same applies to ECMWF which was defined in L50-51 but was not used again.

Please indicate the period used for the analysis (year_start – year_end) or the length of time series used in your analysis.

L126-143: Please explain why you do correlation for JJASON. The Atlantic Niño and AMM are known to have different seasonality with Atlantic Niño peaking in summer and AMM peaking in spring (e.g., see Foltz et al., 2019). These modes are related at least during some periods (e.g., Murtugudde et al., 2001; Foltz and McPhaden, 2010; Martín-Rey et al., 2019).

Also discussion of the Atlantic Niño/Niño 34 correlation need to be improved. It is not enough to say that the correlation is small, you may need to say whether or not it is statistically significant or indicate the p-value. Even if there is no correlation, the ENSO-Atlantic Niño discussion need to be framed in the right perspective. Papers have shown this connection from the Pacific point of view (e.g., Tokinaga et al., 2019) and from Atlantic point of view (e.g., Rodríguez-Fonseca et al., 2009), there is a growing interest tropical basins interactions, rather a total lack of connection you seem to imply here.

L137: It is hard to read 40°W referred to in Fig. 2a.

L136-143: You may need to explain that you have changed the sign of Niño 34 index so that the result here refers to negative phase of ENSO (La Niña).

Figs. S1 and S3: "...post-satellite period" is not used in main manuscript.

L146-164: Recent studies show that the ITCZ/convection and precipitation associated with the Atlantic occurs north of the equator (Richter et al., 2017; Nnamchi et al., 2021). This come out

here with increased precipitation over the southern West Africa (Guinea Coast) and the surrounding ocean region.

The Atlantic Niño does not enhance the West African monsoon, and actually suppress it. For instance Vallès-Casanova et al. (2020) in their Fig. 2c shows that the monsoon region or the Sahel is negatively correlated with the Atlantic Niño which is positively correlated with the rainfall over the coastal region south of 10N. The monsoon is not well correlated with rainfall in this southern coastal region (e.g., see Fig. 1a of Nnamchi and Li, 2011). The West African monsoon is short-lived (about 3 months in July-September) whereas the analysis here is based on JJASON.

L244-270: The specific detailed on how the partial regression was formulated is needed in the Methods section, and discuss the uncertainty/confidence limits.

References:

- Foltz GR et al (2019) The Tropical Atlantic Observing System. *Front. Mar. Sci.* 6:206. doi: 10.3389/fmars.2019.00206
- Foltz GR, McPhaden MJ (2010) Interaction between the Atlantic meridional and Niño modes. *Geophys Res Lett* 37:L18604. <https://doi.org/10.1029/2010GL044001>
- Martín-Rey, M., Lazar, A. Is the boreal spring tropical Atlantic variability a precursor of the Equatorial Mode?. *Clim Dyn* 53, 2339–2353 (2019). <https://doi.org/10.1007/s00382-019-04851-9>
- Nnamchi, H.C., Latif, M., Keenlyside, N.S. et al. Diabatic heating governs the seasonality of the Atlantic Niño. *Nat Commun* 12, 376 (2021). <https://doi.org/10.1038/s41467-020-20452-1>
- Nnamchi, H. C., & Li, J. (2011). Influence of the South Atlantic Ocean Dipole on West African Summer Precipitation, *Journal of Climate*, 24(4), 1184-1197.
- Richter, I., Xie, SP., Morioka, Y. et al. Phase locking of equatorial Atlantic variability through the seasonal migration of the ITCZ. *Clim Dyn* 48, 3615–3629 (2017). <https://doi.org/10.1007/s00382-016-3289-y>
- Rodríguez-Fonseca, B., Polo, I., García-Serrano, J., Losada, T., Mohino, E., Mechoso, C. R., and Kucharski, F. (2009), Are Atlantic Niños enhancing Pacific ENSO events in recent decades? *Geophys. Res. Lett.*, 36, L20705, doi:10.1029/2009GL040048.
- Tokinaga, H., Richter, I., & Kosaka, Y. (2019). ENSO Influence on the Atlantic Niño, Revisited: Multi-Year versus Single-Year ENSO Events, *Journal of Climate*, 32(14), 4585-4600.
- Vallès-Casanova, I., Lee, S.-K., Foltz, G. R. & Pelegrí, J. L. On the spatiotemporal diversity of Atlantic Niño and associated rainfall variability over West Africa and South America. *Geophys. Res. Lett.* 47, e2020GL087108 (2020).

Reviewer #2 (Remarks to the Author):

The Atlantic Niño refers to a mode of sea surface temperature variability in the tropical Atlantic that peaks in the boreal summer. It has been linked to precipitation variability over West Africa and South America. This paper advances the notion that the Atlantic El Niño also impacts tropical cyclone activity over the eastern North Atlantic. The paper suggests that this is accomplished by a strengthening of the Atlantic Intertropical convergence zone (ITCZ), enhancing easterly waves, and low-level cyclonic vorticity over the eastern North Atlantic

Strengths: Interesting results that appear to be novel, Noteworthy result is the potential for the

Atlantic ENSO to impact Atlantic tropical cyclogenesis over and above the known effects of ENSO and other climate modes. This has the potential to improve seasonal hurricane forecasting.

Weaknesses: Some issues related to the method and interpretation as listed below. Additional explanation (but not necessarily additional analysis) is needed to make this paper publishable.

Lines 100-101: This line states the hypothesis that guides the work: "... we hypothesize that Atlantic Niño/Niña modulates Atlantic TC activity primarily through its impact on atmospheric convection over the deep tropical eastern North Atlantic and West Africa and the associated changes in atmospheric circulation and AEW activity in those regions."

Why qualify the hypothesis with the word primarily? Is there a secondary pathway?

Fig 2 shading ought to be improved. Please choose colors that are color-blind-friendly for the general audience. I had trouble seeing the differences in the shading.

Line 147: From what is generally found in past studies, during Atlantic ENSO, rainfall is reduced in the Sahel but enhanced along the Guinea coast. You use the phrase "enhancing the West African summer monsoon rainfall and circulation" which does not make that dipole-like precipitation response.

Line 152: Either use the acronym TNA or "Tropical North Atlantic" consistently in the text.

Line 153: What about the enhanced wind shear right over the eastern Atlantic - where the Cape Verde hurricanes form? You seem to be focussing on the shear that is over the central tropical Atlantic (west of 40 W). Yet, your conclusions emphasize the impact of the Atlantic El Niño on Cape Verde storms.

Line 159: What does fast teleconnection mean?

Fig 3: Please provide more information on how panels (c)-(e) were produced. Provide the regression equations. In particular, what does sum mean in panel 3? As you state earlier, the two Niño indices (Atlantic and Pacific) are not orthogonal. Are you treating them to be approximately orthogonal in panel 3(e)? Some of this confusion may be owing to the lack of clear discussion of this in the methods section.

Also, switching the threshold P-value value to yield 90% confidence versus 95% for other analyses (e.g., panel b) gives the appearance of cherry-picking and violating the spirit of significance testing. This is done in Fig 1 as well.

Also, for other figures where you choose 90% as the confidence threshold, are you comfortable with this choice? What was the rationale behind this number versus say 95% or 99%?

I should also add that statistical significance may be overrated in the sense that just because something is (is not) statistically significant (not significant) may not necessarily mean that it has (does not have) an important effect. Physically based reasoning may be more useful than significance testing.

See, for example, <https://www.nature.com/articles/d41586-019-00857-9>

Line 216: You state: "Therefore, it is more likely that Atlantic Niño/Niña modulates the locations of TC genesis and tracks while the overall seasonal Atlantic TC activity is largely determined by ENSO and AMM."

This is a very general statement and seems reasonable at face value. This is further emphasized by your schematic in Fig 4 where the reader gets the impression that there is an east-west shift in the TC genesis locations depending on the phase of the Atlantic El Nino. This also gives the impression that the basin-wide numbers are the same. However, can you point to any of your results that specifically back up this speculation? For instance, Fig 1 (b,c) shows a nearly single signed TC genesis anomaly in the tropical Atlantic main development region (MDR). Admittedly, the sign is opposite over the Florida coast but does that compensate for the effect within the region south of 20 N (the MDR)?

Signed: Anantha Aiyyer, North Carolina State University

Reviewer #1 (Remarks to the Author):

Review of “Increase in Cape Verde hurricanes during Atlantic Niño”

The study uses observational data sets 1948-2021 to show that the Atlantic Niño events are linked to tropical cyclones and major hurricanes that originate around Cape Verde off the West African Coast. The authors argue that the Atlantic Niños can be linked to stronger the Atlantic inter-tropical convergence zone and enhanced African easterly wave activities and low-level cyclonic vorticity across the deep tropical eastern North Atlantic, the region where the hurricanes originate. In my assessment, this is a novel and timely analysis that is worth publishing in Nature Communications. The major concern I have is that the conclusions are based largely on partial regressions (of the Atlantic Niño, Atlantic meridional mode and El Niño indices) and more details are needed about this method and the robustness of the results. Are the findings/conclusions here sustained using methods such as composite analysis that simply show the years when the Atlantic Niño occurs alone or together with the other modes? I have a few specific comments below.

We greatly appreciate the insightful and detailed comments. We have revised the manuscript by carefully addressing the reviewer’s comments and suggestions. As suggested, we carried out a composite analysis to support our findings, which were largely based on partial regression analysis. Since the number of cases when Atlantic Niño/Niña occurs alone or together with another specific mode (e.g., La Niña) is relatively limited, we increased the sample size for the composite analysis by lowering the criteria for each index to $0.5 \times$ standard deviations. We also added a detailed description of the partial regression method in the revised manuscript. By addressing these and other key points raised by the reviewer, we believe that the revised manuscript is much improved. Our replies (in blue color) are shown below for each of the general and specific comments.

As shown in Fig. R1a, the TC genesis and track density composites during Atlantic Niño under neutral ENSO conditions are largely consistent with those derived from the partial regression analysis (Fig. 2a in the manuscript). In particular, TC genesis is increased over the eastern tropical North Atlantic (east of $\sim 40^\circ\text{W}$) and decreased over the western North Atlantic between 20°N and 30°N . A nearly opposite pattern is shown during Atlantic Niña under neutral ENSO conditions (Fig. R1b).

Similarly, the TC genesis composites when La Niña and Atlantic Niño/Niña occurred simultaneously (Fig. R1c-d) are consistent with those derived from the partial regression analysis (Fig. S5a-b). Specifically, TC genesis increases over the eastern tropical North Atlantic (east of $\sim 40^\circ\text{W}$) when La Niña and Atlantic Niño occurred simultaneously, but slightly decreases (or cancels out) when La Niña and Atlantic Niña occurred simultaneously, supporting our main conclusion that Atlantic Niño has a greater influence on TC genesis over the eastern tropical North Atlantic than La Niña (L221-223, Fig. S5a-b).

The impacts of Atlantic Niño/Niña on TC genesis during positive AMM are also clearly shown in the composite analysis (Fig. R1e-f). For example, when Atlantic Niño occurs during a positive AMM phase, TC genesis increases almost everywhere in the tropical North Atlantic. On the other hand, when Atlantic Niña occurs during a positive AMM phase, TC genesis east of 40°W slightly decreases (or is canceled out). This is consistent with the partial regression analysis (L223-233, Fig. S5c-d). Therefore, we conclude that the composite analyses support our results from the partial regression analysis (Fig. S5). The composite analysis is included in the supporting information. By addressing these,

we modified sentences to “These can be clearly seen in both partial regression and composite analysis results (Fig. S5 and S6) (L233-234).”

Fig. R1. Spatial patterns of anomalous TC genesis (shaded, per year) and track density (black contours, intervals of 0.4 TCs per year) composites during (a) Atlantic Niño only and (b) Atlantic Niña only cases. Purple dots indicate where TC genesis anomalies are significant above the 90% confidence level based on a Student’s t-test. TC genesis and track density are spatially smoothed to aid visual comparison. (c) and (d) are spatial patterns of anomalous TC genesis (shaded, per year) composites during La Niña & Atlantic Niño and La Niña & Atlantic Niña, respectively. (e) and (f) are the same as (c) and (d) but for TC genesis during positive AMM & Atlantic Niño, and positive AMM & Atlantic Niña. Note that Atlantic Niño/Niña, El Niño/La Niña, and (+)/(-) AMM are identified when their corresponding indices exceed $0.5 \times$ standard deviation during JJASON or are less than $-0.5 \times$ standard deviation.

L26-28: Not clear what “they” refers to, also when the hurricane season is.

We replaced “they” with “those climate modes” and specified hurricane season (June-November) in L26-28.

L54-62: The argumentation here is a bit hard to follow; it reads confusing and perhaps even confusing.

The fast tropical teleconnection mechanism was discussed by Horel and Wallace (1981), Yulaeva and Wallace (1994), and Chiang and Sobel (2002). It describes equatorial atmospheric Kelvin waves that produce a global average warming of the tropical troposphere during El Niño. The teleconnected tropospheric warming over the tropical Atlantic tends to increase atmospheric static stability. Additionally, the meridional tropospheric temperature gradient within and across the edge of the tropics is enhanced. This in turn directly increases the vertical wind shear over the Atlantic main development region (MDR, 10°N–20°N and 85°W–15°W), via the thermal wind relationship (e.g., Lee et al., 2011; Larson et al., 2012).

To clarify “the fast tropical teleconnection”, we added “(via atmospheric Kelvin waves)” in L55. Additionally, these sentences are further revised to make them easier to follow: “For instance, during La Niña, the negative phase of El Niño - Southern Oscillation (ENSO), cold equatorial Pacific sea surface temperature anomalies (SSTAs) produce a fast tropical teleconnection (via atmospheric Kelvin waves) that decreases the atmospheric static stability and vertical wind shear over the TC main development region (MDR) and thus increase Atlantic TC activity. The positive phase of the Atlantic Meridional Mode (AMM), characterized by warm SSTAs and low-level westerly wind anomalies over the tropical North Atlantic, also decreases the atmospheric static stability and vertical wind shear over the MDR, and thus increases Atlantic TC genesis. Conversely, El Niño, the positive phase of ENSO, and the negative phase of AMM both tend to increase the atmospheric static stability and vertical wind shear over the MDR and thus suppress Atlantic TC genesis. (L53-63)”

L109-114: I think that what is needed here is a summary of what was done and the most important findings.

These sentences are now revised to “The main objectives of this study are to investigate further the links between Atlantic Niño/Niña and seasonal Atlantic TC activity and explore the associated physical mechanisms by analyzing observational and reanalysis datasets. We show the critical role of Atlantic Niño/Niña in the development of major Atlantic hurricanes in the deep tropics near the Cape Verde islands. (L110-113)”

L117-125: GoM, CBN etc. These unfamiliar acronyms make reading the manuscript difficult, especially where they are not used many time. The same applies to ECMWF which was defined in L50-51 but was not used again.

Several acronyms, including GoM, CBN, TNA and ECMWF, are now removed in the revised manuscript.

Please indicate the period used for the analysis (year_start – year_end) or the length of time series used in your analysis.

We added the period of our analysis in the revised manuscript (L117 and L173), figure captions, and supplementary.

L126-143: Please explain why you do correlation for JJASON. The Atlantic Niño and AMM are known to have different seasonality with Atlantic Niño peaking in summer and AMM peaking in spring (e.g., see Foltz et al., 2019). These modes are related at least during some periods (e.g., Murtugudde et al., 2001; Foltz and McPhaden, 2010; Martín-Rey et al., 2019).

We agree with the reviewer. However, our objective in this study is not to explore the physical link between Atlantic Niño/Niña and AMM or the mechanism of Atlantic Niño/Niña development. We are mainly interested in the impact of Atlantic Niño/Niña on Atlantic TC activity and comparing that to the impact of AMM (or ENSO). Therefore, we computed the instantaneous correlation between Atlantic Niño and AMM for the entire Atlantic hurricane season of June-November (JJASON), which is virtually zero ($r=-0.03$).

To address the reviewer's point about the time-lagged relationship between AMM and Atlantic Niño/Niña in some years, we modified these sentences to “Two leading modes of SSTA variability in the tropical Atlantic are Atlantic Niño/Niña and AMM. Although some Atlantic Niño/Niña events are preceded by AMM events in boreal spring (e.g., Murtugudde et al., 2001; Foltz and McPhaden, 2010; Martín-Rey et al., 2019; Vallès-Casanova et al., 2020), the correlation between ATL3 and AMM indices (Methods) during the Atlantic hurricane season (JJASON) is virtually zero ($r=-0.03$). (L127-130)”

Also discussion of the Atlantic Niño/Niño 34 correlation need to be improved. It is not enough to say that the correlation is small, you may need to say whether or not it is statistically significant or indicate the p-value. Even if there is no correlation, the ENSO-Atlantic Niño discussion need to be framed in the right perspective. Papers have shown this connection from the Pacific point of view (e.g., Tokinaga et al., 2019) and from Atlantic point of view (e.g., Rodríguez-Fonseca et al., 2009), there is a growing interest tropical basins interactions, rather a total lack of connection you seem to imply here.

In the revised manuscript, we indicate statistical significance of the correlation between ATL3 and NINO34. The lead-lag correlation between ATL3 and NINO34 (e.g., Tokinaga et al., 2019; Vallès-Casanova et al. 2020; Rodríguez-Fonseca et al., 2009; Münnich and Neelin, 2005) is also addressed by adding “However, while the correlation between ATL3 and NIÑO3.4 indices in JJASON is statistically insignificant based on a two-tailed Student's t-test ($r = -0.15$), some Atlantic Niño/Niña events are directly forced by ENSO in boreal winter and spring, and Atlantic Niño/Niña may influence ENSO evolution in boreal summer and fall (e.g., Münnich and Neelin, 2005; Rodríguez-Fonseca et al., 2009; Tokinaga et al., 2019; Vallès-Casanova et al. 2020). Therefore, to properly address the potential interbasin interactions and their impact on Atlantic TC activity, it is necessary to separate the impact of Atlantic Niño/Niña on Atlantic TCs from that of ENSO. (L130-136)”

L137: It is hard to read 40°W referred to in Fig. 2a.

We changed the x-axis interval in Fig 2a and other spatial patterns figures.

L136-143: You may need to explain that you have changed the sign of Nino 34 index so that the result here refers to negative phase of ENSO (La Nina).

We added “Note that the sign of NIÑO3.4 is reversed in Fig. 2b to help visual comparison. (L137-138)”.

Figs. S1 and S3: “...post-satellite period” is not used in main manuscript.

We defined the post-satellite period as a period of 1979-2021 (L144) and added it to the revised manuscript (L186, L190, L218 and L260) and supplementary (Fig. S1 and Fig. S3).

L146-164: Recent studies show that the ITCZ/convection and precipitation associated with the Atlantic occurs north of the equator (Richter et al., 2017; Nnamchi et al., 2021). This come out here with increased precipitation over the southern West Africa (Guinea Coast) and the surrounding ocean region. The Atlantic Niño does not enhance the West African monsoon, and actually suppress it. For instance Vallès-Casanova et al. (2020) in their Fig. 2c shows that the monsoon region or the Sahel is negatively correlated with the Atlantic Niño which is positively correlated with the rainfall over the coastal region south of 10N. The monsoon is not well correlated with rainfall in this southern coastal region (e.g., see Fig. 1a of Nnamchi and Li, 2011). The West African monsoon is short-lived (about 3 months in July-September) whereas the analysis here is based on JJASON.

Thank you very much for this point. As the reviewer explained, Atlantic Niño enhances the sub-Sahel region rainfall (0-10N; e.g., Richter et al., 2017; Vallès-Casanova et al. 2020; Nnamchi and Li, 2011) but often suppresses rainfall over the Sahel region (e.g., Nnamchi et al., 2021). To address the reviewer's concern, we changed "West African monsoon" to "sub-Sahel West African rainfall" in the revised version. We think that "sub-Sahel West African rainfall" fits better with our proposed mechanism and also with previous studies. For example, Grodsky et al. (2003) suggested that sub-Sahel rainfall and AEWs have a positive correlation. Thorncroft and Hodge (2001) showed that AEWs that form over the west sub-Sahel region show a positive correlation with tropical cyclone activity at interannual time scales. Richter et al. (2017), Nnamchi et al. (2021), and Nnamchi and Li (2011) are now referenced in the revised manuscript (L151).

L244-270: The specific detailed on how the partial regression was formulated is needed in the Methods section, and discuss the uncertainty/confidence limits.

Thank you for pointing out this. We added a detailed description of the partial regression method in the revised Method section (L279-294):

"In order to objectively separate the impact of Atlantic Niño/Niña on Atlantic TC from that of ENSO, we computed partial regressions of Atlantic TC genesis and track density onto ATL3 and NIÑO3.4 indices as follows

$$\begin{aligned} TC_{ENSO_ATL3}(t, x, y) &= \beta_1(x, y) \cdot NINO34(t) + \beta_2(x, y) \cdot ATL3(t) \\ TC_{ENSO}(t, x, y) &= \beta_1(x, y) \cdot NINO34(t) \\ TC_{ATL3}(t, x, y) &= \beta_2(x, y) \cdot ATL3(t) \end{aligned}$$

where the TC_{NINO34_ATL3} is the reconstructed TC variable using partial regressions (Fig. 3e). β_1 is the partial regression coefficient of NINO34. It represents TC change per unit change in NINO34 while ATL3 is held constant. Similarly, β_2 is the partial regression coefficient of ATL3, representing TC change per unit change in ATL3 while NINO3.4 is held constant. NINO34 and ATL3 are normalized. TC_{NINO34} is the reconstructed TC variable using only the partial regression coefficient of NINO34. TC_{ATL3} is same as TC_{NINO34} but for using only the partial regression coefficient of ATL3. Using the reconstructed TC genesis and track density, we computed the correlations between observed and reconstructed TC genesis/track density as shown in Fig. 3c-e. The significance test conducted in this study is based on a standard two-tailed Student's t-test."

In supplementary Fig. S5, we show the linear summation and subtraction of partial regression of tropical cyclone genesis and track density using ATL3 and NINO3.4 indices to explore the compound impact of ATL3 and NINO3.4 as well as the impact of ATL3 and AMM. Additionally, the partial regression assumes that Atlantic TC activity is only a function of NINO3.4 (or AMM) and ATL3 while the actual TCs are potentially influenced by other natural variability and anthropogenic forcing.

References:

- Horel, J. D., and J. M. Wallace (1981), Planetary-scale atmospheric phenomena associated with the Southern Oscillation, *Mon. Weather Rev.*, 109, 813–829, doi:10.1175/1520-0493(1981)109<0813:PSAPAW>2.0.CO;2.
- Yulaeva, E., and J. M. Wallace (1994), The signature of ENSO in global temperature and precipitation fields derived from the microwave sounding unit, *J. Clim.*, 7, 1719–1736, doi:10.1175/1520-0442(1994)007<1719:TSEOIG>2.0.CO;2.
- Chiang, J. C. H., and A. H. Sobel (2002), Tropical tropospheric temperature variations caused by ENSO and their influence on the remote tropical climate, *J. Clim.*, 15, 2616–2631, doi:10.1175/1520-0442(2002)015<2616:TTVCB>2.0.CO;2.
- Lee, S.-K., D. B. Enfield, and C. Wang (2011), Future impact of differential inter-basin ocean warming on Atlantic hurricanes, *J. Clim.*, 24, 1264–1275, doi:10.1175/2010JCLI3883.1.
- Larson, S., Lee, S.-K., Wang, C., Chung, E.-S., & Enfield, D. B. Impacts of non-canonical El Niño patterns on Atlantic hurricane activity. *Geophys. Res. Lett.* 39, (2012).
- Foltz, G., et al. The tropical Atlantic observing system. *Front. Mar. Sci.* 6, 206. <https://doi.org/10.3389/fmars.2019.00206> (2019).
- Foltz G.R. & McPhaden M. J. Interaction between the Atlantic meridional and Niño modes. *Geophys Res Lett*, 37, L18604. <https://doi.org/10.1029/2010GL044001> (2010).
- Martín-Rey, M. & Lazar, A. Is the boreal spring tropical Atlantic variability a precursor of the Equatorial Mode? *Clim. Dyn.*, 53, 2339–2353, <https://doi.org/10.1007/s00382-019-04851-9> (2019).
- Nnamchi, H. C., & Li, J. Influence of the South Atlantic Ocean Dipole on West African Summer Precipitation, *J. Clim.*, 24, 1184–1197 (2011).
- Nnamchi, H.C., Latif, M., Keenlyside N. S., Kjellsson J. & Richter I. Diabatic heating governs the seasonality of the Atlantic Niño. *Nat. Commun.*, 12, 376. <https://doi.org/10.1038/s41467-020-20452-1>(2021).
- Richter, I., Xie, S.P., Morioka, Y. Doi. T. Taguchi B. & Behera S. Phase locking of equatorial Atlantic variability through the seasonal migration of the ITCZ. *Clim. Dyn.*, 48, 3615–3629. <https://doi.org/10.1007/s00382-016-3289-y> (2017).
- Rodríguez-Fonseca, B., Polo, I., García-Serrano, J., Losada, T., Mohino, E., Mechoso, C. R., & Kucharski, F. Are Atlantic Niños enhancing Pacific ENSO events in recent decades? *Geophys. Res. Lett.*, 36, L20705, doi:10.1029/2009GL040048 (2009).
- Tokenaga, H., Richter, I., & Kosaka, Y. ENSO Influence on the Atlantic Niño, Revisited: Multi-Year versus Single-Year ENSO Events *J. Clim.* 32, 4585–4600., <https://journals.ametsoc.org/view/journals/clim/32/14/jcli-d-18-0683.1.xml> (2019).
- Vallès-Casanova, I., Lee, S.-K., Foltz, G. R., & Pelegrí, J. L. On the spatiotemporal diversity of Atlantic Niño and associated rainfall variability over West Africa and South America. *Geophys. Res. Lett.* 47, e2020GL087108. <https://doi.org/10.1029/2020GL087108> (2020).
- Münnich, M., & Neelin J. D. Seasonal influence of ENSO on the Atlantic ITCZ and equatorial South America. *Geophys. Res. Lett.*, 32, L21709, <https://doi.org/10.1029/2005GL023900> (2005).
- Grodsky, S. A., Carton, J. A., & Nigam, S. Near surface westerly wind jet in the Atlantic ITCZ, *Geophys. Res. Lett.*, 30, doi:10.1029/2003GL017867 (2003).

Thorncroft, C., & Hodges K. African Easterly Wave variability and its relationship to Atlantic Tropical Cyclone Activity, *J. Clim.* 14, 1166– 1179 (2001).

Reviewer #2 (Remarks to the Author):

The Atlantic Niño refers to a mode of sea surface temperature variability in the tropical Atlantic that peaks in the boreal summer. It has been linked to precipitation variability over west Africa and South America. This paper advances the notion that the Atlantic El Niño also impacts tropical cyclone activity over the eastern North Atlantic. The paper suggests that this is accomplished by a strengthening of the Atlantic Intertropical convergence zone (ITCZ), enhancing easterly waves, and low-level cyclonic vorticity over the eastern North Atlantic

Strengths: Interesting results that appear to be novel, Noteworthy result is the potential for the Atlantic ENSO to impact Atlantic tropical cyclogenesis over and above the known effects of ENSO and other climate modes. This has the potential to improve seasonal hurricane forecasting.

Weaknesses: Some issues related to the method and interpretation as listed below. Additional explanation (but not necessarily additional analysis) is needed to make this paper publishable.

We greatly appreciate the insightful and detailed comments and suggestions. We have revised the manuscript based on the reviewer's comments and suggestions. In particular, we improved the method section in the revised manuscript by adding a detailed description of the partial regression method. Our replies (in blue color) are shown below for each of the general and specific comments.

Lines 100-101: This line states the hypothesis that guides the work: "... we hypothesize that Atlantic Niño/Niña modulates Atlantic TC activity primarily through its impact on atmospheric convection over the deep tropical eastern North Atlantic and West Africa and the associated changes in atmospheric circulation and AEW activity in those regions." Why qualify the hypothesis with the word primarily? Is there a secondary pathway?

Thank you for the reviewer's comment. We deleted "primarily" in the revised manuscript (L102).

Fig 2 shading ought to be improved. Please choose colors that are color-blind-friendly for the general audience. I had trouble seeing the differences in the shading.

We appreciate the reviewer's suggestion. We changed the color bars in Fig 2.

Line 147: From what is generally found in past studies, during Atlantic ENSO, rainfall is reduced in the Sahel but enhanced along the Guinea coast. You use the phrase "enhancing the West African summer monsoon rainfall and circulation" which does not make that dipole-like precipitation response.

Thank you very much for this point. As the reviewer explained, Atlantic Niño enhances the sub-Saharan region rainfall (0-10N; e.g., Richter et al., 2017; Vallès-Casanova et al. 2020; Nnamchi and Li, 2011) but often suppresses rainfall over the Sahel region (e.g., Nnamchi et al., 2021). To address the reviewer's concern, we changed "West African monsoon" to "sub-Saharan West African rainfall" in the revised version. We think that "sub-Saharan West African rainfall" fits better with our proposed mechanism and also with previous studies. For example, Grodsky et al. (2003) suggested that sub-Saharan rainfall and AEWs have a positive correlation. Thorncroft and Hodge (2001) showed that AEWs that form over the west sub-Saharan region show a positive correlation with tropical cyclone activity at interannual time scales.

Line 152: Either use the acronym TNA or "Tropical North Atlantic" consistently in the text.

We removed TNA in the revised manuscript.

Line 153: What about the enhanced wind shear right over the eastern Atlantic - where the Cape Verde hurricanes form? You seem to be focussing on the shear that is over the central tropical Atlantic (west of 40 W). Yet, your conclusions emphasize the impact of the Atlantic El Niño on Cape Verde storms.

The vertical wind shear is slightly enhanced in the far eastern tropical North Atlantic during Atlantic Niño as well as during La Niña (Fig. 2e & f). However, the increased wind shear is centered slightly north of the equator where Atlantic TCs rarely form. So, it should not have much impact on TC genesis. This is also consistent with earlier studies. For instance, Landsea et al (1998) showed strong positive vertical wind shear in the deep tropics during the extremely active 1995 hurricane season (Fig. 4a in Landsea et al., 1998). The 1995 hurricane season is also one of the strong Atlantic Niño years, which is consistent with our findings.

Line 159: What does fast teleconnection mean?

The fast tropical teleconnection mechanism was discussed by Horel and Wallace (1981), Yulaeva and Wallace (1994), and Chiang and Sobel (2002). It describes equatorial atmospheric Kelvin waves that produce a global average warming of the tropical troposphere during El Niño. The teleconnected tropospheric warming over the tropical Atlantic tends to increase atmospheric static stability. Additionally, the meridional tropospheric temperature gradient within and across the edge of the tropics is enhanced. This in turn directly increases the vertical wind shear over the Atlantic main development region (MDR, 10°N–20°N and 85°W–15°W), via the thermal wind relationship (e.g., Lee et al., 2011; Larson et al., 2012).

To clarify “the fast tropical teleconnection”, we added “(via atmospheric Kelvin waves)” in L55. Additionally, these sentences are further revised to make them easier to follow: “For instance, during La Niña, the negative phase of El Niño - Southern Oscillation (ENSO), cold equatorial Pacific sea surface temperature anomalies (SSTAs) produce a fast tropical teleconnection (via atmospheric Kelvin waves) that decreases the atmospheric static stability and vertical wind shear over the TC main development region (MDR) and thus increase Atlantic TC activity. The positive phase of the Atlantic Meridional Mode (AMM), characterized by warm SSTAs and low-level westerly wind anomalies over the tropical North Atlantic, also decreases the atmospheric static stability and vertical wind shear over the MDR, and thus increases Atlantic TC genesis. Conversely, El Niño, the positive phase of ENSO, and the negative phase of AMM both tend to increase the atmospheric static stability and vertical wind shear over the MDR and thus suppress Atlantic TC genesis. (L53-63)”

Fig 3: Please provide more information on how panels (c)-(e) were produced. Provide the regression equations. In particular, what does sum mean in panel 3? As you state earlier, the two Niño indices (Atlantic and Pacific) are not orthogonal. Are you treating them to be approximately orthogonal in panel 3(e)? Some of this confusion may be owing to the lack of clear discussion of this in the methods section.

Also, switching the threshold P-value value to yield 90% confidence versus 95% for other analyses (e.g., panel b) gives the appearance of cherry-picking and violating the spirit of significance testing. This is done in Fig 1 as well.

Also, for other figures where you choose 90% as the confidence threshold, are you comfortable with this choice? What was the rationale behind this number versus say 95% or 99%?

I should also add that statistical significance may be overrated in the sense that just because something is (is not) statistically significant (not significant) may not necessarily mean that it has (does not have) an important effect. Physically based reasoning may be more useful than significance testing.

See, for example, <https://www.nature.com/articles/d41586-019-00857-9>

We appreciate these suggestions and comments. We added a detailed description of the partial regression method in the revised Method section (L279-294): “In order to objectively separate the impact of Atlantic Niño/Niña on Atlantic TC from that of ENSO, we computed partial regressions of Atlantic TC genesis and track density onto ATL3 and NIÑO3.4 indices as follows

$$\begin{aligned}TC_{ENSO_ATL3}(t, x, y) &= \beta_1(x, y) \cdot NINO34(t) + \beta_2(x, y) \cdot ATL3(t) \\TC_{ENSO}(t, x, y) &= \beta_1(x, y) \cdot NINO34(t) \\TC_{ATL3}(t, x, y) &= \beta_2(x, y) \cdot ATL3(t)\end{aligned}$$

where the TC_{NINO34_ATL3} is the reconstructed TC variable using partial regressions (Fig. 3e). β_1 is the partial regression coefficient of NINO34. It represents TC change per unit change in NINO34 while ATL3 is held constant. Similarly, β_2 is the partial regression coefficient of ATL3 representing TC change per unit change in ATL3 while NINO3.4 is held constant. NINO34 and ATL3 are normalized. TC_{NINO34} is the reconstructed TC variable using only the partial regression coefficient of NINO34. TC_{ATL3} is same as TC_{NINO34} but for using only the partial regression coefficient of ATL3. Using the reconstructed TC genesis and track density, we computed the correlation between observed and reconstructed TC genesis/track density as shown in Fig. 3c-e. The significance test conducted in this study is based on a standard two-tailed Student's t-test.”

We have also updated all of the significant tests by using a 95% significant confidence level instead of a 90%. We have obtained consistent results and conclusions with the updated significant confidence level. We agree with the reviewer that a significant test does not guarantee a causal relationship between two variables.

Line 216: You state: "Therefore, it is more likely that Atlantic Niño/Niña modulates the locations of TC genesis and tracks while the overall seasonal Atlantic TC activity is largely determined by ENSO and AMM."

This is a very general statement and seems reasonable at face value. This is further emphasized by your schematic in Fig 4 where the reader gets the impression that there is an east-west shift in the TC genesis locations depending on the phase of the Atlantic El Nino. This also gives the impression that the basin-wide numbers are the same. However, can you point to any of your results that specifically back up this speculation? For instance, Fig 1 (b,c) shows a nearly single signed TC genesis anomaly in the tropical Atlantic main development region (MDR). Admittedly, the sign is opposite over the Florida coast but does that compensate for the effect within the region south of 20 N (the MDR)?

We agree with the reviewer. To address this point, we now indicate three hurricanes for La Niña (Fig. 4b) and only two for Atlantic Niño (Fig. 4a) in the revised manuscript.

Signed: Anantha Aiyyer, North Carolina State University

References:

- Richter, I., Xie, S.P., Morioka, Y. Doi. T. Taguchi B. & Behera S. Phase locking of equatorial Atlantic variability through the seasonal migration of the ITCZ. *Clim. Dyn.*, 48, 3615–3629. <https://doi.org/10.1007/s00382-016-3289-y> (2017).
- Rodríguez-Fonseca, B., Polo, I., García-Serrano, J., Losada, T., Mohino, E., Mechoso, C. R., & Kucharski, F. Are Atlantic Niños enhancing Pacific ENSO events in recent decades? *Geophys. Res. Lett.*, 36, L20705, doi:10.1029/2009GL040048 (2009).
- Vallès-Casanova, I., Lee, S.-K., Foltz, G. R., & Pelegrí, J. L. On the spatiotemporal diversity of Atlantic Niño and associated rainfall variability over West Africa and South America. *Geophys. Res. Lett.* 47, e2020GL087108. <https://doi.org/10.1029/2020GL087108> (2020).
- Nnamchi, H. C., & Li, J. Influence of the South Atlantic Ocean Dipole on West African Summer Precipitation, *J. Clim.*, 24, 1184-1197 (2011).
- Nnamchi, H.C., Latif, M., Keenlyside N. S., Kjellsson J. & Richter I. Diabatic heating governs the seasonality of the Atlantic Niño. *Nat. Commun.*, 12, 376. <https://doi.org/10.1038/s41467-020-20452-1>(2021).
- Grodsky, S. A., Carton, J. A., & Nigam, S. Near surface westerly wind jet in the Atlantic ITCZ, *Geophys. Res. Lett.*, 30, doi:10.1029/2003GL017867 (2003).
- Thorncroft, C., & Hodges K. African Easterly Wave variability and its relationship to Atlantic Tropical Cyclone Activity, *J. Clim.* 14, 1166– 1179 (2001).
- Landsea, C. W., Bell G. D., Gray W. M. & Goldenberg S. B. The extremely active 1995 Atlantic hurricane season: Environmental conditions and verification of seasonal forecasts. *Mon. Wea. Rev.*, 126, 1174–1193 (1998).
- Horel, J. D., and J. M. Wallace (1981), Planetary-scale atmospheric phenomena associated with the Southern Oscillation, *Mon. Weather Rev.*, 109, 813–829, doi:10.1175/1520-0493(1981)109<0813:PSAPAW>2.0.CO;2.
- Yulaeva, E., and J. M. Wallace (1994), The signature of ENSO in global temperature and precipitation fields derived from the microwave sounding unit, *J. Clim.*, 7, 1719–1736, doi:10.1175/1520-0442(1994)007<1719:TSOEIG>2.0.CO;2.
- Chiang, J. C. H., and A. H. Sobel (2002), Tropical tropospheric temperature variations caused by ENSO and their influence on the remote tropical climate, *J. Clim.*, 15, 2616–2631, doi:10.1175/1520-0442(2002)015<2616:TTVCB>2.0.CO;2.
- Lee, S.-K., D. B. Enfield, and C. Wang (2011), Future impact of differential inter-basin ocean warming on Atlantic hurricanes, *J. Clim.*, 24, 1264–1275, doi:10.1175/2010JCLI3883.1.
- Larson, S., Lee, S.-K., Wang, C., Chung, E.-S., & Enfield, D. B. Impacts of non-canonical El Niño patterns on Atlantic hurricane activity. *Geophys. Res. Lett.* 39, (2012).

REVIEWERS' COMMENTS

Reviewer #1 (Remarks to the Author):

Review of "Increase in Cape Verde hurricanes during Atlantic Niño" by Dongmin Kim et al.,

The author addressed my comments on the previous version of the manuscript that bordered on the robustness of the analysis and discussion of previous literature. I do not have any further comments and would look forward to reading the published version of the manuscript.

Reviewer #2 (Remarks to the Author):

The authors have made reasonable changes in response to the comments made by both reviewers. I feel that the paper can be accepted at this stage.